# Factors Associated with Vaginal/Cesarean Birth Attitudes among Medical Students

**DOI:** 10.3390/healthcare10030571

**Published:** 2022-03-18

**Authors:** Anna Michalik, Agnieszka Czerwińska-Osipiak, Anna Szablewska, Michalina Pracowity, Jolanta Olszewska

**Affiliations:** Department of Obstetrical and Gynaecological Nursing, Medical University of Gdansk, Debinki 7, 80-211 Gdansk, Poland; aczerwinska@gumed.edu.pl (A.C.-O.); anna.szablewska@gumed.edu.pl (A.S.); m.pracowity@gumed.edu.pl (M.P.); jolanta.olszewska@gumed.edu.pl (J.O.)

**Keywords:** medical students, attitudes, cesarean, vaginal birth

## Abstract

Background: Polish perinatal care is facing a high, ever-increasing cesarean section (CS) rate that is currently at 43%. Crucially, reports have revealed that the attitudes, experiences, and skills of clinicians directly contribute to this elevated CS rate. Methods: This cross-sectional study, which included 748 Polish medical students, aimed to identify medical students’ attitudes regarding birth methods. A descriptive questionnaire was distributed via the academic email addresses of surveyed medical students. Group comparisons were performed using Welch’s *t*-test for continuous data or a Chi-squared test for categorical data. We also used the Mann–Whitney U test and Kruskal–Wallis H test. Results: Midwifery students (96.2%) were the most unified group of students, with most agreeing that VB (vaginal birth) presents a safer option for women at low risk for VB-related complications vs. cesarean section. Of Medical Faculty students, 68% believed that fewer complications typically occur during vaginal birth than during CS. Students in their final vs. initial years of study furthermore considered VB more beneficial for women than CS. Conclusions: An important factor identified at the individual clinician level is the presence of leadership and executive support. For medical students, we can interpret this as support from their trainers and supervisors.

## 1. Introduction

The use of cesarean section (CS) is increasing worldwide, with a current global rate of 21.1%. Researchers estimate the continuation of this tendency across all global regions [1,2]. Access to safe cesarean delivery represents a fundamental aspect of modern perinatal care. CS can effectively prevent maternal and perinatal mortality and morbidity. However, the distribution of CS across different populations does not meet all the recommendations, and medically justified, unmet needs coexist with overuse of the procedure [2,3,4]. The increasing number of CS cases constitutes one of the most frequently discussed topics in perinatal care worldwide, particularly as it applies to pregnant women from groups at low risk from vaginal birth (VB)-related complications. As with any surgery, CS is associated with short- and long-term complications that can extend many years beyond the delivery at hand, potentially affecting the health of the woman, her child, and future pregnancies. Addressing in detail the potential risks and benefits associated with CS procedures is vital and of particular importance for those situations in which CS is not indicated for the benefit of the neonate [5,6]. 

In the absence of effective interventions to reduce CS frequency, populations are facing and will continue to face a complex scenario involving morbidity and mortality associated with unmet needs, unsafe provision of CS, and the concomitant overuse of this surgical procedure, which drains resources and risks avoidable morbidity and mortality [2,7]. The World Health Organization (WHO) asserts that “every effort should be made to provide cesarean sections to women in need, rather than striving to achieve a specific rate,” showing that medical justification should underly decision-making processes surrounding CS, as opposed to population-level recommendations [4]. Data-driven discussions are essential in evaluating and implementing effective interventions to reduce the number of CS cases.

Poland is a European Union member state with a population of nearly 38 million, representing the largest population among Central and Eastern European countries [8]. Polish perinatal care faces a high and still-increasing CS rate; in Poland, the current CS rate is 43% [9], much higher than the European average of 27% [10,11]. The highest European CS rate was found in Romania (46.9%) and the lowest in the Netherlands (14.9%) [2]. Polish perinatal care has been observed to trend in two conflicting directions: the medicalization of childbirth, often promoted by obstetricians, and de-medicalization, which is often supported by women and midwives. This tension may contribute to confusing messaging and a lack of consistency in approaches to childbirth that together may complicate decision-making for women about their preferred birth route. In addition, epidural availability in Poland is insufficient (only 40% of birthing women having access to this procedure). Interestingly, the highest CB rates are reported in the regions of Poland where epidural access is the most limited. Recent Polish reports show that the rate of medical interventions (birth induction, augmentation, episiotomy, amniocentesis, non-spontaneous pushing techniques, overuse of intrapartum cardiotocographic fetal monitoring) is continually rising [12,13,14]. Wide variation in intervention rates between populations with similar demographic and health profile characteristics raises doubt about the appropriateness of implemented procedures and suggests that different practices and attitudes concerning the extent to which evidence-based clinical guidelines should be followed exist [15,16]. A logical conclusion from the above concerns is that the practical education of Polish medical students in the area of perinatal care is based on a medicalized, interventional system. Students more often observe interventions during birth rather than seeing a birth follow its physiological course. What students see and experience during practical training shapes their future practice, including in pregnancy- and birth-related topics.

There is a direct connection between the knowledge, experience, and attitudes of medical professionals and their procedure distribution and variations implemented [15]. Crucially, reports have suggested that the attitudes, experiences, and skills of clinicians have been recognized as an important driver of CS rates. Moreover, the motivations and opinions of obstetricians often have a stronger effect on CS rates than clinical patterns or scientific evidence of benefit [16,17,18].

The objective of this study was to identify medical students’ opinions and attitudes concerning vaginal birth and cesarean section. We posit that these attitudes reflect theoretical and practical education on perinatal care and can help identify areas for modification.

## 2. Materials and Methods

### 2.1. Study Design and Research Group

This exploratory, cross-sectional study was conducted within a group of Polish medical students between January and May of 2021. All respondents were informed of the study aims and the planned manner of publication for the results, and all provided their voluntary consent to participate. A descriptive questionnaire, which allowed for the collection of quantitative data, was administered online. The questionnaire was piloted in a group of 10 students to verify that the questions would be easily understood by respondents. We distributed the research tool via the personal, academic email addresses of the medical students. A Google Form questionnaire, along with a description of the study, was sent to the medical students following consent from their universities. This research had a nationwide reach, and we focused on recruiting a diversity of respondents regarding field of study.

The eligibility screening was an integral part of the questionnaire. Inclusion criteria for the study included: current medical student status (field of study must include education concerning perinatal care) and a completed survey returned from a university email address. To determine the size of the representative group of the finite population of medical students (total number of medical students at the universities in the 2021/22 academic year is 6196) [8], a commonly available sample size calculator was used. With an assumed confidence level (95%) and a maximum error of 5%, the minimum sample size was set at 362 respondents. Finally, of 765 survey attempts, 17 were excluded (incomplete), leaving a total sample size of *N* = 748.

### 2.2. Research Tools

#### Introductory Section

The final version of the survey included 40 questions divided into two sections:

Section A: demographic characteristics (age, gender, urban/rural habitation, field of study, year of education).

Section B: Medical students’ opinions about CS and VB, including knowledge of Polish proportions of VB:CS and WHO recommendations in this area, opinions concerning which method of birth (VB vs. CS) is safer for low-risk pregnant woman, and whether every woman should have the right to choose between VB and CS. In this section, we also investigated knowledge concerning the birth and postpartum period (e.g., factors that can intensify the pain experience, such as temperature in the room, number of people in the room, level of information concerning ongoing situation, and light intensity).

We measured medical students’ opinions using standardized tools:

To assess medical students’ attitudes and opinions, we used a 5-point Likert-type scale, with 1 = “Definitely no”, 2 = “No”, 3 = “I don’t know”, 4 = “Yes”, and 5 = “Definitely yes”.

To assess pain experienced by women during VB and CS, which is subjective, we used the most frequently recommended pain intensity assessment tool, the Numerical Rating Scale (NRS), in which 0 represents no pain and 10 represents the strongest possible pain. The Numerical Rating Scale was combined with a Visual Analogue Scale (VAS) in the form of a 100 mm segment, the left end of which represents no pain while the right end represents the strongest possible pain. The VAS is one of the most frequently used tools in Poland to describe pain subjectively experienced by a patient [19]. Both scales are used in daily practice to assess birth pain [20].

Statistical analysis of the results was carried out using IBM SPSS 23 packages and R (R Core Team, 2018). Qualitative variables were presented by means of counts and percentages, and the quantitative variable was characterized by means of an arithmetic mean and standard deviation. The significance of differences between more than two groups was checked by the Kruskal–Wallis test (when significant differences were obtained, Bonferroni’s post hoc tests were used), and between two groups by the Student’s *t*-test for independent and dependent samples, and Mann–Whitney’s U test. Chi square tests were used for qualitative variables, and *p* ≤ 0.05 was used as the significance level in all calculations. 

The study protocols were approved by the Independent Bioethics Committee for Scientific Research at the Medical University of Gdańsk, Poland.

## 3. Results

### 3.1. Demographic Characteristics of the Sampled Students

The studied group consisted of 748 medical students, aged 18–46 years (mean 22.22), studying in different medical fields. For all students included, their scope of practice included (on different levels) pregnant women and women of reproductive age. The largest group of respondents comprised Medicine Faculty students (31.6%). Detailed demographic characteristics are presented in Table 1. 

### 3.2. Overall Knowledge of Polish Perinatal Care and Opinions Concerning Vaginal Birth (VB) and Cesarean Birth (CB)

In Table 2, we present overall knowledge of Polish perinatal care and opinions concerning vaginal birth (VB) and cesarean birth (CB).

Overall, the results revealed that Polish medical students have appropriate knowledge of perinatal care fundamentals. The largest proportion of the studied group (55.3%) indicated that currently, between 41% and 60% of children in Poland are born vaginally. Furthermore, 46.8% of respondents did not believe that the proportion of cesarean sections carried out in Poland aligns with WHO recommendations regarding perinatal care.

### 3.3. Student Opinions Regarding VB and CS

Figure 1 presents students’ answers to the question “Do you think that VB is safer for low risk for VB-related complications women, than CS?” according to their field of study.

Results showed that 68.9% of students considered vaginal birth safer than delivery via cesarean section. According to the majority of respondents (83.6%), women’s recovery from VB is speedier than from CS, and 62.2% of participants thought that every woman should be entitled to the choice of delivery method and the possibility of selecting a cesarean section in any scenario, regardless of medical indications. Field and year of study are factors that appear to influence student attitudes. The results showed that 96.2% of Midwifery students agreed that vaginal childbirth is a safer option than cesarean section (χ2 (9) = 78.89, *p* < 0.001) for low risk for VB-related complications women. Students studying Midwifery, Nursing, Clinical Nutrition, and Medical Faculty were more likely to believe that women’s recovery after VB is faster than post-CS, a finding that is statistically significant (χ2 (9) = 69.88, *p* < 0.001). Most Health Psychology (92.9%) and Electroradiology (88%) students thought that every woman should have the freedom of choice regarding birth method. The group most opposed to women having a choice between VB and CS regardless of circumstances was Midwifery students (68%); in their view, medical indications are crucial in decisions involving CS (χ2 (9) = 93.77, *p* < 0.001). We additionally demonstrated that respondents who significantly more often did not agree with the opinion that VB is safer than CS also believed that every woman should be included in childbirth decision-making under any circumstances and informed of the availability of CS as a delivery option (χ2 (1) = 49.35, *p* < 0.001). Midwifery students most frequently believed that a lack of trust in their own bodies, as well as an absence of their own feminine acceptance, influenced women’s decisions concerning delivery method (χ2 (9) = 24.84, *p* = 0.003). Analysis factoring concerning respondents’ year of study showed that those in years III, IV, V, and VI had statistically significantly different beliefs to those in years I and II, viewing VB as safer and more beneficial, particularly for women at low risk for VB-related complications (χ2 (5) = 16.85, *p* = 0.005). Furthermore, students in years I and II were significantly more likely to believe that every woman should have the freedom of choice regarding childbirth method (χ2 (5) = 24.60, *p* < 0.001). Students in years III to VI were significantly more likely to indicate that women recover faster after VB than CS (χ2 (5) = 28.47, *p* < 0.001). (Figure 2)

We also found that students living in urban vs. rural areas were more likely to support women having VB vs. choosing CS (χ2 (5) = 8.17, *p* = 0.043). In addition, students whose self-assessed knowledge of pregnancy and childbirth was high were significantly more likely to believe that women should not have the ability to choose a cesarean delivery in any given case. Those who rated themselves as having only minimal or sufficient knowledge, however, were more likely to hold the opinion that every woman should have the right to choose their birth method (χ2 (3) = 60.46, *p* < 0.001). A further component related to differences between VB and CS that we analyzed was students’ perception of pain during childbirth and 24 h postpartum. A paired samples t-test confirmed that students attending perinatal care courses were more likely to believe that pain levels are higher during VB as opposed to CB (t(747) = 49.72, *p* < 0.001), whereas pain experienced 24 h postpartum, according to respondents, is higher after CS (t(747) = 3.15, *p* < 0.05). 

### 3.4. Students’ Opinions about Pain Levels after VB and CS

To investigate factors influencing respondents’ opinions, a series of statistical analyses were carried out. A Kruskal–Wallis H test revealed that Midwifery and Clinical Nutrition students rated pain during VB the lowest, significantly lower than students in other fields (χ2 (9) = 42.28, *p* < 0.001). In contrast, the levels of pain during CS were rated lowest by students studying Medicine, Physiotherapy, and Emergency Medicine (χ2 (9) = 21.12, *p* < 0.05). Midwifery students rated pain 24 h post-VB the lowest (χ2 (9) = 136.08, *p* < 0.001) and pain post-CS the highest (χ2 (9) = 57.95, *p* < 0.001). Considering year of study, results showed that pain levels 24 h post-VB were viewed as significantly higher by respondents in years I, II, III, and IV compared to those in years V and VI (χ2 (5) = 25.53, *p* < 0.001). 

Differences were also detectable in the context of postpartum recovery. The Mann–Whitney U test demonstrated that respondents who believed that women regain their strength and wellbeing more quickly after CS vs. VB also rated the levels of pain during VB as higher, as opposed to students who believed that women’s recoveries progress better after VB (*Z* = 2.83, *p* < 0.05).

Students’ opinions concerning birth method pain levels (VAS scoring) are presented in Table 3. 

Pain levels 24 h post-CS were rated higher by students who also believed that women recover faster from VB (*Z* = 7.72, *p* < 0.001). It is also worth noting that students who expressed the opinion that every woman should be able to exert influence over the birth method and have the freedom to choose a cesarean section regardless of circumstances rated significantly higher pain levels on the VAS scale during VB (*Z* = −1.97, *p* < 0.05) and 24 h postpartum (*Z* = −7.61, *p* < 0.001) (as determined by Mann–Whitney U test). Taking into account student aspirations and the diversification of fields of study, we determined that students whose scope of professional practice would not include child-birthing women assessed pain levels as being higher 24 h post-VB than post-CS (Z = 2.40, *p <* 0.05). In contrast, students whose profession would include child-birthing women estimated pain levels as being higher 24 h post-CS than post-VB (Z = 2.26, *p* < 0.05). 

## 4. Discussion

Recent data indicate that 43% of all children born in Poland are born through CS, though the underlying cause of this unusually high CS rate remains unknown [9]. A high rate of CS poses a significant problem for the medical community in many countries and is even considered a global epidemic [21]. Several studies focused on possible causes of high and still-increasing CS rates have found that variability in CS rates is driven by differences in clinician knowledge, values, and beliefs, as well as cultural and social factors and the medicalization of pregnancy and births [17,18,22,23]. We thus hypothesize that attitudes and knowledge of medical students—future clinicians—concerning VB and CS is the effect of all aforementioned factors but additionally, and perhaps primarily, of standardized theoretical and practical education as the foundation for their future practices. Therefore, after delineating factors that may contribute to the unusually high CS rate in Poland, we asked medical students about their attitudes towards VB and CS. Students entering into medical practice are confronted with the paradox of learning through curriculum that the optimization of the number of CSs performed among the population of healthy pregnant women is one of the greatest challenges facing modern obstetric practice, yet they are then becoming part of a highly interventional system where the CS rate remains one of the highest in Europe [9,10,11,12,13,14,15,16]. A lack of unified guidelines and unwarranted variation in procedure distribution were also identified as factors contributing to the rise in CS preference and intervention implementation among clinicians. This trend often stems from the belief that a controlled and planned course of CS is relatively safer than the uncontrollable nature of VB [24].

In this study, we were interested in medical students’ opinions and attitudes about VB vs. CS. Midwifery students (96.2%) were the most unified group of students, with most agreeing that VB is a safer option than CS for women at low risk of VB-related complications. There are some visible differences in medical students’ opinions about this issue. The possible explanation is how general, populational trends (mass and social media included) influence the received education in the perinatal care area. The wider the program of perinatal care during the study field, the more students that hold the opinion that VB is safer and more beneficial for healthy pregnant women. It can also address some gaps in education, that should drive the further discussion. Of Medical Faculty students, 68% believed that fewer complications typically occur during vaginal birth vs. CS. This group, which includes future obstetricians, was less convinced in this area than Emergency Medicine, Clinical Nutrition, and Nursing students. We hypothesize that this result is the effect of the fact that the obstetrician takes responsibility for the course of VB (and the decision about the CS). As we mentioned, this trend stems from the convenience that the planned and controlled CS is also safer for the practitioner. Additionally, in the scope of education of future physicians is the intervention, when pathology occurs. This can affect the interventions overuse. The impact of education on medical students’ attitudes is evident, given that Polish medical students’ opinions about VB and CS can change over their years of study: more students in their final years assessed VB as a safer, more beneficial option compared to CS. They also thought that women recover faster after VB than after cesarean delivery, another opinion potentially stemming from years of study. Students studying Midwifery, Nursing, Clinical Nutrition, and Medical Faculty more often held the belief that women’s recovery post-VB is faster than post-CS. Relevant results from other countries vary—we can find data confirming these opinions (that VB is the preferred mode of delivery in a healthy future pregnancy among young students [25,26]), but also contradictory reports, in which medical students stated that they would prefer a CS for the birth of their own child [27]. However, in groups of medical students, in which CS is the favorable mode of birth, a significantly greater proportion of last-year students opted for CS compared to entry-year students [27,28,29]. This distinction may be the crux of the matter, that preferences for CS among medical students reflect a wide range of factors, including education, both practical and theoretical, and cultural and social background.

The opportunity for a pregnant woman to choose her birth route, including open access to CS available “on request” without medical indication, has been identified as a factor contributing to elevated cesarean rates [17,30]. Observations from everyday practice and various reports indicate that women who are convinced that CS is better for them exist across cultural settings, and these women express a strong preference for CS for their mode of delivery. In Polish practice, it is not recommended to perform cesareans without medical justification. We investigated this issue through Polish medical students’ opinions and found that throughout the course of medical studies, attitudes about giving women the choice in birth options “on request” are also changing: we found that students in their final years are more likely to formulate the opinion that women should not have the opportunity to choose compared to students in their beginning years. We think that this is the effect of education and knowledge about Polish recommendations: in accordance with them, it is not recommended/allowed to undergo CS “on request” without medical justification. Respondents whose scope of practice is not directly connected with perinatal practice—Health Psychology and Electroradiology students—were convinced that every woman should have the freedom of choice regarding their birth method. The group most strongly opposed to women having a cesarean without medical indication was that of Midwifery students. In addition, we found that the respondents who significantly more often did not agree that VB is safer than CS also believed that every woman should have the opportunity to choose CS as a birth route, despite medical indications. Students who self-assessed their knowledge of pregnancy and childbirth as being high believed significantly more often that women should not have the choice to opt for a cesarean delivery in any given case.

Reports have identified a lack of pain alleviation during VB and birth-related pain fear as significant variables associated with elevated CS rates [4,18,31]. Moreover, in some students’ opinions, pain associated with vaginal delivery was a reason to opt for CS [27,32,33]. We analyzed the relationship between the differences between VB and CS and student perceptions of pain both during childbirth and 24 h postpartum. Polish medical students held the opinion that pain levels are higher during VB vs. CS, whereas pain experienced 24 h postpartum (post-VB and post-CS) is higher after CS. Midwifery and Clinical Nutrition students rated the pain during VB the lowest and significantly lower than students from other fields. In contrast, pain levels during CS were rated lowest by students undertaking Medicine, Physiotherapy, and Emergency Medicine. Respondents typically believed that recovery from VB is shorter than from CS.

A strategy to reduce the trend of performing unnecessary CS is the promotion of VB, including preparing women for pregnancy and birth and offering constant support during the birth itself [18]. Equally important is supporting students in their personal development. The important factor at the individual clinician level is having leadership and executive support [32,34]. For medical students, we can interpret this as support from their trainers and supervisors [35]. A disconnect between theoretical recommendations and practical observations in hospital settings can lead to lower confidence, elevated stress levels, and decisional conflicts among medical students. The effect of these seems to constitute a factor that corresponds with elevated CS rates.

Data from a recent quantitative and qualitative review suggest that despite the global rise in CS rates over the last few decades, vaginal birth remains the preferred mode of delivery for a healthy future pregnancy among young students. However, the perception of high-risk vaginal delivery, a family history of CS births, and fear of labor all relate to CS preferences [25,33]. Preferences for CS and obstetric interventions correlate with attitudes and beliefs related to cultural, social, and psychological factors. As knowledge backgrounds vary, it should be emphasized that educational programs through the media and social media can predispose young people to specific decisions regarding childbirth while increasing their awareness of the various options of childbirth in terms of evidence-based practices. The authors of this review advise that fostering a positive attitude towards a given behavior can lead to adoption of the behavior. 

This is the first Polish study about medical students’ attitudes about VB and CB and their possible impact on their future practice in the perinatal care area. We designed exploratory research, based on a self-constructed questionnaire. A limitation of this study was the little research in this field, which required us to establish a benchmark in relation to the research question and study design. Our results can be a foundation for another, vertical research, with more detailed questions and focused on variation between different study fields. Despite the fact that we reached multiple groups of respondents, it would be more valuable to design a national, randomized control trial. The knowledge about the change of attitudes towards VB and CB in the course of medical students’ education (first year vs. final year) would be beneficial to include. 

## 5. Conclusions

In Poland, where the CB rate is 43%, our research contributes to the fundamental knowledge about the described phenomenon. There is a visible difference in VB and CS attitudes among medical students from various fields, which is not in line with current recommendations and evidence-based medical knowledge. An important factor identified at the individual clinician level is the presence of leadership and executive support. For medical students, we can interpret this as support from their trainers and supervisors.

## Figures and Tables

**Figure 1 healthcare-10-00571-f001:**
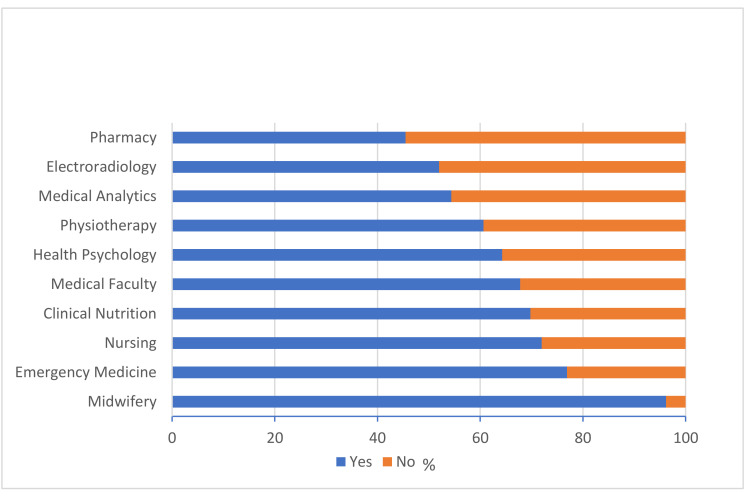
Students’ answers to the question: “Do you think that VB is safer than C-section?” Organized according to field of study.

**Figure 2 healthcare-10-00571-f002:**
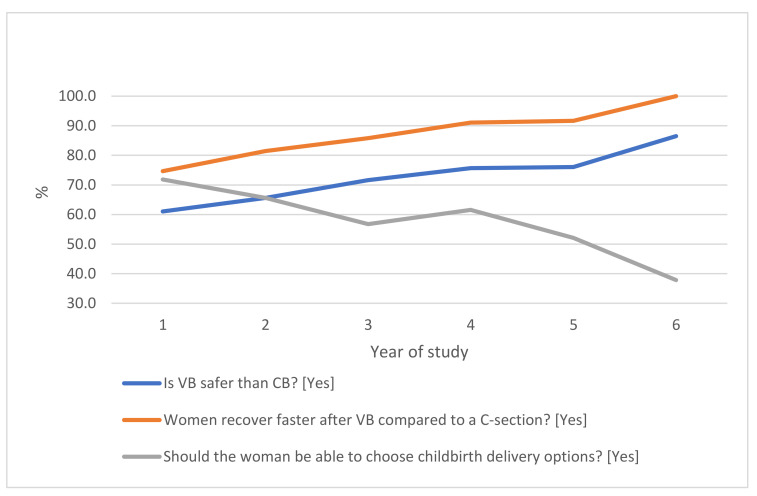
Changes in students’ opinions about VB and CS over years of study.

**Table 1 healthcare-10-00571-t001:** Characteristics of the group/general information about the respondents.

Characteristics	*N*	*%*
Age, mean ± SD (standard deviation) 22.22 ± 2.61	748	
Place of residence		
Rural	176	23.5
Urban	572	76.4
Sex		
Female	646	86.4
Male	102	13.6
Field of study		
Midwifery	131	17.5
Nursing	82	11
Medical Faculty	236	31.6
Clinical Nutrition	43	5.7
Physiotherapy	61	8.2
Pharmacy	88	1.8
Emergency Medicine	13	1.7
Medical Analytics	46	6.1
Health Psychology	14	1.9
Electroradiology	25	3.3
Public Health	5	0.7
Dentistry	4	0.5
Year of study		
I	213	28.5
II	183	24.5
III	141	18.9
IV	78	10.4
V	96	12.8
VI	37	4.9
The future professional practice include the care of pregnant and birthing women		
Yes	488	65.2
No	260	34.8
Your biological capacity of having children (in your knowledge and opinion)		
Yes	734	98.1
No	14	1.9

**Table 2 healthcare-10-00571-t002:** Overall opinions and knowledge concerning vaginal/cesarean birth.

Statements	*N*	*%*
Most births in Poland are		
VB	617	82.5
CB	131	17.5
The current proportion of VB in Poland is		
0–20%	4	0.5
21–40%	114	15.2
41–60%	414	55.3
More than 60%	216	28.9
The current proportion of CB in Poland is		
0–20%	62	8.3
21–40%	460	61.5
41–60%	188	25.1
More than 60%	38	5.1
The Polish CB rate follows WHO recommendations		
Yes	85	11.4
No	350	46.8
I don’t know	313	41.8
Type of birth recommended for pregnant women from groups at low risk for vaginal birth (VB)-related complications		
VB	716	95.7
CB	32	4.3
VB is safer and more beneficial for the mother and baby compared with CB		
Yes	515	68.9
No	233	31.1
Every woman should have the right to opt for CB in any situation, independent of existing medical indications		
Yes	625	83.6
No	123	16.4
The lack of trust in one’s own body and lack of acceptance of one’s femininity influence the decision to choose childbirth delivery options		
Yes	628	84
No	120	16
The rate of your own knowledge of pregnancy and childbirth		
Very good	89	11.9
Good	220	29.4
Sufficient	214	28.6
Small	219	29.3
Lack of any knowledge	6	0.8

**Table 3 healthcare-10-00571-t003:** Opinions about birth method pain levels (VAS scoring).

VAS Score for Birth Pain Intensity	N	Min	Max	M	SD
VAS score for VB pain level	748	4	10	9.03	1.04
VAS score for CS pain level	748	0	10	4.11	2.65
VAS score for pain level 24 h after VB	748	0	10	5.41	2.19
VAS score for pain level 24 h after CS	748	0	10	5.76	2.23

## Data Availability

Data available on request due to privacy/ethical restrictions.

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
