# Peer review of "Factors Associated with Vaginal/Cesarean Birth Attitudes among Medical Students"

_healthcare, 2022, doi:10.3390/healthcare10030571_

Round 1
Reviewer 1 Report
Abstract
L#20 Explain what is VB ---Vaginal birth?
Introduction
Compare the prevalence of CS in Poland with the recommended rate of CS by the WHO. Also, highlight the factors associated with the high prevalence of CS in Poland.
Apart from the medical profession, there are several factors that influence CS. The authors only cited one study reference 15 to strengthen their argument related to "vaginal/cesarean birth attitudes among medical students". I would suggest building your argument on "the attitude of medical students" and the CS rate. Moreover, also discuss why the authors did not interview the physicians?
Methods
Discuss the non-response rate. How many students were targeted and how many filled the questionnaire?
Line 121-123. Explain logic to employ the mentioned statical analysis.
Add a section on variable and measurement in your method section. This section should discuss all your variables and measures.
How did the authors calculate sample size?
Run the strobe guideline for reporting the result of cross-sectional studies.
https://www.equator-network.org/wp-content/uploads/2015/10/STROBE_checklist_v4_cross-sectional.pdf
Results:
Table 01. Last two questions
The question statement should not be similar to the study question. The author can re-write "future professional practice includes the care of
pregnant and birthing women". Moreover, follow this rule in table 02 as well.
Discussion:
The discussion related to figure 01 "Student answers to question: “Do you think that VB is safer than C-section?”; organized according to the field of study" can further discuss why there is a gap in knowledge within medical professionals?
Moreover, also discuss why senior medical students are not in favor of choosing childbirth delivery options by women?
Add strength and limitation section
Add brief study conclusions.
Author Response
We sincerely thank you for the review and feedback. The paper has been analysed in terms of the questions asked. We are hoping that the changes introduced into the paper will allow us to better present our concept of the issue.
We are enclosing our answers and comments in Word file below.

Reviewer 2 Report
The article entitled "Factors associated with vaginal/cesarean birth attitudes among medical students" deals with a very important topic.
Indeed, the percentage of CS without medical indications is alarmingly high in Poland.
Educating future medical staff in the field of VB benefits and CS risk seems to be crucial for changing their personal beliefs and attitudes towards taking up professional behaviours for the health of both the mother and the newborn.
I would like to thank the authors for taking up this topic.
The article is a well-written scientific report.
I only miss information about the effect and sample size. Whether and how they were estimated. Please provide suitable information.
To improve the manuscript, I also recommend supplementing the discussion with the limitations of the study and implications for further research exploration of this issue.
Author Response
We sincerely thank you for the review and feedback. The paper has been analysed in terms of the questions asked. We are hoping that the changes introduced into the paper will allow us to better present our concept of the issue.
We are enclosing our answers and comments below:
Rev. 2: "I only miss information about the effect and sample size. Whether and how they were estimated. Please provide suitable information."
Answer:
- We added: To determine the size of the representative group of the finite population of medical students (total number of medical students at the universities in the 2021/22 academic year is 6196) [8] a commonly available sample size calculator was used. With an assumed confidence level (95%) and a maximum error of 5% the minimum sample size is set at 362 responds. Finally, of 765 survey attempts, 17 were excluded (incompleted), leaving a total sample size of N = 748.
Line 104 – 109
Rev. 2: "To improve the manuscript, I also recommend supplementing the discussion with the limitations of the study and implications for further research exploration of this issue."
Answer: The paragraph about limitations of the study and further explorations in this issue has been introduced at the end of the discussion. Line 354 - 363